# Clustering a Union of Linear Subspaces via Matrix Factorization and Innovation Search

**Mostafa Rahmani**

Amazon Prime Video, Seattle WA
mostrahm@amazon.com

## Abstract

This paper focuses on the Matrix Factorization based Clustering (MFC) method which is one of the few closed-form algorithms for the subspace clustering algorithm. Despite being simple, closed-form, and computation-efficient, MFC can outperform the other sophisticated subspace clustering methods in many challenging scenarios. We reveal the connection between MFC and the Innovation Pursuit (iPursuit) algorithm which was shown to be able to outperform the other spectral clustering based methods with a notable margin especially when the span of clusters are close. A novel theoretical study is presented which sheds light on the key performance factors of both algorithms (MFC/iPursuit) and it is shown that both algorithms can be robust to notable intersections between the span of clusters. Importantly, in contrast to the theoretical guarantees of other algorithms which emphasized on the distance between the subspaces as the key performance factor and without making the innovation assumption, it is shown that the performance of MFC/iPursuit mainly depends on the distance between the innovative components of the clusters.

## 1 INTRODUCTION

When data points lie in a single linear manifold, conventional techniques such as Principal Component Analysis (PCA) can be efficiently used to find the underlying low-dimensional structure [Zhang and Lerman, 2014, Lerman et al., 2015]. However, in many applications, the data points may be originating from multiple independent sources and a union of manifolds can better model the data [Vidal, 2011]. The subspace clustering problem is defined on how to learn these low dimensional manifolds when they are linear subspaces [Heckel and Bölcskei, 2013, Elhamifar and Vidal,

2013, Tsakiris and Vidal, 2017, Rahmani and Atia, 2017a, Peng et al., 2016, Lu et al., 2013, Feng et al., 2014, Patel et al., 2013, Wang et al., 2013, Li et al., 2021, Lu et al., 2013, Wang and Xu, 2016, You et al., 2016, Ji et al., 2017, Zhang et al., 2018, Klys et al., 2018, Peng et al., 2016, Menon et al., 2020, Jiang et al., 2018, Lipor et al., 2021] in a completely unsupervised way.

**Summary of contributions:** This paper focuses on analyzing two subspace clustering algorithms: Matrix Factorization based Clustering (MFC) and Innovation Pursuit (iPursuit). First we reveal the underlying connection between them and the presented analysis shows why they can notably outperform other spectral clustering based methods in the challenging scenarios. The main contributions of this work can be summarized as follows.

● It is shown that iPursuit is equivalent to MFC if we alter its $\ell_1$-norm based cost function into a quadratic cost function and importantly, all the presented theoretical results are applicable to both algorithms.

● To the best of our knowledge, this paper presents the first comprehensive analysis of MFC/iPursuit algorithms and the presented analysis is not based on the restrictive innovation assumption used in [Rahmani and Atia, 2017a,b]. The MFC/iPursuit algorithms are analyzed and we establish deterministic and probabilistic sufficient conditions which guarantee that the computed adjacency matrix by MFC/iPursuit satisfies a defined quality requirement. Importantly, it is shown that in contrast to most of other clustering algorithms whose performance depend on the distance between the subspaces, the performance of MFC/iPursuit mainly depends on the distance between the innovative components of the clusters. Accordingly, even if the span of clusters intersect heavily, MFC/iPursuit can still provably satisfy the performance requirement.

**Notation and Definitions:** Given a matrix $\mathbf{A}$, $\|\mathbf{A}\|$ denotes its spectral norm, $\|\mathbf{A}\|_F$ denotes its Frobenius norm, and $\|\mathbf{A}\|_{p,1} = \sum_i \|\mathbf{a}_i\|_p$ where $\mathbf{a}_i$ denotes the $i^{th}$ column of $\mathbf{A}$ and $\mathbf{a}^i$ denotes the $i^{th}$ row of $\mathbf{A}$. For a vector $\mathbf{a}$, $\|\mathbf{a}\|_p$

*Accepted for the 38$^{th}$ Conference on Uncertainty in Artificial Intelligence* (UAI 2022).

denotes its $\ell_p$-norm, $\mathbf{a}(i)$ denotes its $i^{\text{th}}$ element, and $\mathbf{a}[i:k]$ contains the elements of $\mathbf{a}$ whose indexes are from $i$ to $k$. The elements of matrix $\mathbf{Y} = |\mathbf{X}|$ are equal to the absolute value of the elements of matrix $\mathbf{X}$. The subspace $\mathcal{U}^\perp$ is the complement of $\mathcal{U}$. $\mathbb{S}^{M_1-1}$ indicates the unit $\ell_2$-norm sphere in $\mathbb{R}^{M_1}$. It is assumed that data matrix $\mathbf{D} \in \mathbb{R}^{M_1 \times M_2}$ can be represented as $\mathbf{D} = \mathbf{U}\Sigma\mathbf{V}^T$ where $\mathbf{U} \in \mathbb{R}^{M_1 \times r_d}$ is the matrix of left singular vectors, the diagonal matrix $\Sigma \in \mathbb{R}^{r_d \times r_d}$ contains the non-zero singular values, the columns of $\mathbf{V} \in \mathbb{R}^{M_2 \times r_d}$ are equal to the right singular vectors, $r_d$ is the rank of $\mathbf{D}$, $M_2$ is the number of data points, and $M_1$ is the dimension of ambient space. The subspace $\mathcal{S} = \oplus_{i=1}^m \mathcal{S}_i$ is equal to the direct sum of subspaces $\{\mathcal{S}_i\}_{i=1}^m$ and $\dim(\mathcal{S})$ denotes the dimension of $\mathcal{S}$. Two adjacency matrices $\mathbf{A} \in \mathbb{R}^{M_2 \times M_2}$ and $\mathbf{B} \in \mathbb{R}^{M_2 \times M_2}$ are said to be equivalent when $\frac{\mathbf{a}_i}{\|\mathbf{a}_i\|_1} = \frac{\mathbf{b}_i}{\|\mathbf{b}_i\|_1}$ holds for all $1 \le i \le M_2$. RHS means right hand side and LHS means left hand side.

**Distance between subspaces:** Suppose $\mathbf{U}_1 \in \mathbb{R}^{M_1 \times r}$ and $\mathbf{U}_2 \in \mathbb{R}^{M_1 \times r}$ are orthonormal bases for r-dimensional subspaces $\mathcal{S}_1$ and $\mathcal{S}_2$, respectively. Two different notions are used to express the affinity between two subspaces. One measure is $\|\mathbf{U}_1^T\mathbf{U}_2\|$. However, $\|\mathbf{U}_1^T\mathbf{U}_2\|$ is always equal to 1 when $\dim(\mathcal{S}_1 \cap \mathcal{S}_2) > 0$. The other measure of affinity between two subspaces is

$$\|\mathbf{U}_1^T\mathbf{U}_2\|_\sigma = \sqrt{\frac{\sum_{i=1}^r \cos^2 \theta_i}{r}}$$

where $\{\theta_i\}_{i=1}^r$ are the principal angles between $\mathcal{S}_1$ and $\mathcal{S}_2$ [Soltanolkotabi et al., 2012]. Note that $\|\mathbf{U}_1^T\mathbf{U}_2\|_\sigma = 1$ only when $\mathcal{S}_1 = \mathcal{S}_2$.

## 1.1 DATA MODEL

Data Model 1 provides the details of the presumed model along with definition of the used symbols. To simplify the exposition and the analysis, it is assumed that the dimension of subspaces are equal, the number of data points in different clusters are equal, and a subspace $\mathcal{S}$ is used to define the intersection between the span of clusters.

**Data Model 1.** *The data matrix $\mathbf{D} \in \mathbb{R}^{M_1 \times M_2}$ can be written as $\mathbf{D} = [\mathbf{D}_1, \mathbf{D}_2, ..., \mathbf{D}_m]\mathbf{T}$ where $\mathbf{T} \in \mathbb{R}^{M_2 \times M_2}$ is an unknown permutation matrix. We define $\mathcal{S}_i$ as the column space of $\mathbf{D}_i$ and $\mathcal{S}_i \not\subset \mathcal{S}_j$ and $\mathcal{S}_j \not\subset \mathcal{S}_i$ for any $i \ne j$. The dimension of all subspaces is equal to $r$ and there are $n$ data points in each cluster, i.e., $\mathbf{D}_i \in \mathbb{R}^{M_1 \times n}$. The dimension of the intersection between subspaces is equal to $s$, i.e., $\dim(\cap_{i=1}^m \mathcal{S}_i) = s$ and we define subspace $\mathcal{S} = \cap_{i=1}^m \mathcal{S}_i$. In addition, $\mathcal{S}_i \cap \mathcal{S}_j = \mathcal{S}$ for all $i \ne j$. The orthonormal matrix $\mathbf{U}_i \in \mathbb{R}^{M_1 \times r}$ is a basis for $\mathcal{S}_i$ and it can be written as $\mathbf{U}_i = [\mathbf{S}, \dot{\mathbf{U}}_i]$ where orthonormal matrix $\mathbf{S} \in \mathbb{R}^{M_1 \times s}$ is a basis for $\mathcal{S} = \cap_{i=1}^m \mathcal{S}_i$ and $\dot{\mathbf{U}}_i \in \mathbb{R}^{M \times (r-s)}$ is a basis for $\mathcal{S}_i \cap \mathcal{S}^\perp$. The orthonormal matrix $\dot{\mathbf{U}}_i$ represents the component of $\mathcal{S}_i$ which does not lie in $\mathcal{S}$ and we call $\dot{\mathcal{S}}_i = \text{span}(\dot{\mathbf{U}}_i) = \mathcal{S}_i \cap \mathcal{S}^\perp$ the innovative component of $\mathcal{S}_i$. Each data point $\mathbf{d}_i$ which lies in $\mathcal{S}_{k_i}$ can be represented as*

$$\mathbf{d}_i = \mathbf{S}\alpha_i + \dot{\mathbf{U}}_{k_i}\beta_i, \tag{1}$$

*where $\alpha_i \in \mathbb{R}^s$ and $\beta_i \in \mathbb{R}^{r-s}$.*

In order to represent the association of each data point to its corresponding cluster, we define index $k_i$ such that $\mathbf{d}_i \in \mathcal{S}_{k_i}$. Matrix $\mathbf{D}_{-k}$ includes all the columns of $\mathbf{D}$ except the ones which lie in $\mathcal{S}_k$. Matrices $\dot{\mathbf{D}}_j$ and $\bar{\mathbf{D}}_j$ are defined as $\dot{\mathbf{D}}_j = \dot{\mathbf{U}}_j^T\mathbf{D}_j$ and $\bar{\mathbf{D}}_j = \mathbf{S}^T\mathbf{D}_j$.

---

**Algorithm 1** Data Clustering Using iPursuit

---

**Input.** The input is data matrix $\mathbf{D} \in \mathbb{R}^{M_1 \times M_2}$.

**1. Project data points on $\mathbb{S}^{M-1}$.** Set $\mathbf{d}_i$ equal to $\mathbf{d}_i/\|\mathbf{d}_i\|_2$ for all $1 \le i \le M_2$.

**2. Direction search.** Define $\mathbf{C}^* \in \mathbb{R}^{M_1 \times M_2}$ as optimal point of $\min_{\mathbf{C}} \|\mathbf{C}^T\mathbf{D}\|_1$ subject to $\text{diag}(\mathbf{C}^T\mathbf{D}) = \mathbf{1}$.

**3.** Define adjacency matrix $\mathbf{A} = |\mathbf{C}^T\mathbf{D}|$.

**4.** Normalize the $\ell_1$-norm of each row of $\mathbf{A}$ (i.e., normalize degree to 1) and apply graph preprocessing steps (e.g., sparsifying adjacency matrix $\mathbf{A}$ via keeping few dominant non-zero elements of each row).

**5.** Apply spectral clustering to $\mathbf{A} + \mathbf{A}^T$.

**Output:** The identified clusters.

---

## 2 RELATED WORK

Numerous approaches for subspace clustering were proposed in prior work including statistical-based approaches [Yang et al., 2006, Tipping and Bishop, 1999, Sugaya and Kanatani, 2004, Fischler and Bolles, 1981], spectral clustering based methods [Elhamifar and Vidal, 2013, Liu et al., 2013], the algebraic-geometric approach [Vidal et al., 2005], and iterative methods [Bradley and Mangasarian, 2000]. Much of the recent research work on subspace clustering is focused on spectral clustering [Von Luxburg, 2007] based methods [Dyer et al., 2013, Gao et al., 2015, Elhamifar and Vidal, 2013, Heckel and Bölcskei, 2013, Liu et al., 2013, Rahmani and Atia, 2017c, Soltanolkotabi et al., 2012, Wang et al., 2013, Chen and Lerman, 2009, Park et al., 2014].

The spectral clustering based algorithms are composed of two main steps and they only differ in the first step. First, an adjacency matrix is constructed via finding a neighborhood set for each data point and in the second step, the spectral graph clustering algorithm [Von Luxburg, 2007] is applied to the learned adjacency matrix. For instance, Sparse Subspace Clustering (SSC) [Elhamifar and Vidal, 2013] uses $\ell_1$-minimization to construct a sparse adjacency matrix, Low-Rank Representation (LRR) [Liu et al., 2013] uses nuclear norm minimization to find the adjacency matrix, and the Thresholding based Subspace Clustering (TSC) method [Heckel and Bölcskei, 2013] simply uses the inner-product between the data points to construct the adjacency

matrix. In contrast to TSC which uses inner-product between the data points to construct the adjacency matrix, iPursuit [Rahmani and Atia, 2017a,c] utilized the directions of innovation to measure the similarity between the data points. The Matrix Factorization based Clustering (MFC) method [Kanatani, 2001, Costeira and Kanade, 1998, Boult and Brown, 1991] is a closed-from spectral clustering based method which utilizes the right singular vectors of the data to construct the adjacency matrix.

---

**Algorithm 2** Matrix Factorization based Clustering (MFC)

---

**Input.** The input is data matrix $\mathbf{D} \in \mathbb{R}^{M_1 \times M_2}$.

**1. Project data points on $\mathbb{S}^{M-1}$.** Set $\mathbf{d}_i$ equal to $\mathbf{d}_i/\|\mathbf{d}_i\|_2$ for all $1 \leq i \leq M_2$.

**2. SVD:** Compute $\mathbf{D} = \mathbf{U}\boldsymbol{\Sigma}\mathbf{V}^T$ where the columns of $\mathbf{V} \in \mathbb{R}^{M_2 \times r_d}$ are equal to the right singular vectors.

**3.** Define $\mathbf{A} = |\mathbf{V}\mathbf{V}^T|$.

**4.** Similar to Step 4 in Algorithm 1.

**5.** Similar to Step 5 in Algorithm 1.

**Output:** The identified clusters.

---

## 2.1 A BRIEF OVERVIEW OF IPURSUIT (ALGORITHM 1)

Suppose that data matrix $\mathbf{D}$ follows Data Model 1. If the span of clusters satisfy Assumption 1, then we say that Innovation Assumption holds.

**Assumption 1.** *For each subspace $\mathcal{S}_i$, we have $\mathcal{S}_i \notin \underset{k \neq i}{\oplus} \mathcal{S}_k$.*

Define orthonormal matrix $\mathbf{P}_i$ such that the column-space of $\mathbf{P}$ is equal to $\mathcal{P}_i = \underset{k \neq i}{\oplus} \mathcal{S}_k$. If the innovation assumption holds, then the rank of $(\mathbf{I} - \mathbf{P}_i\mathbf{P}_i^T)\mathbf{U}_i$ is greater than zero and we define $\vec{\mathcal{S}}_i$ as the column-space of $(\mathbf{I} - \mathbf{P}_i\mathbf{P}_i^T)\mathbf{U}_i$. The geometrical idea behind iPursuit is that if we can find a direction in $\vec{\mathcal{S}}_i$, it is orthogonal to all the clusters except $\mathcal{S}_i$ and this fact can be used to distinguish $\mathcal{S}_i$ from the rest of clusters. Specifically, in order to find a direction in $\vec{\mathcal{S}}_{k_i}$ corresponding to each $\mathbf{d}_i$, [Rahmani and Atia, 2017a,c] proposed to find this direction (dubbed the direction of innovation corresponding to $\mathbf{d}_i$) as the optimal point of

$$\min_{\mathbf{c}} \ \|\mathbf{c}^T\mathbf{D}\|_1 \quad \text{subject to} \quad \mathbf{c}^T\mathbf{d}_i = 1 \, . \tag{2}$$

The motivation behind the design of (2) was that the direction of innovation corresponding to $\mathbf{d}_i$ can be computed via looking for a vector which is orthogonal to the maximum number of data points. Although the innovation assumption was used to design iPursuit, in [Rahmani and Atia, 2017c,a] it was numerically shown that it is not essential in the performance of iPursuit.

The authors of [Rahmani and Atia, 2017c,a] presented an analysis of (2) which is limited to a two cluster scenario

and it was based on the Innovation Assumption to prove that the optimal point of (2) lies in $\vec{\mathcal{S}}_{k_i}$. In contrast, the presented theoretical study (a) does not require the innovation assumption, (b) guarantees a completely different requirement, (c) is the first thorough analysis of MFC, (d) reveals the connection between iPursuit and MFC, and importantly (e) it shows the importance of the incoherence between the innovative components.

# 3 ANALYZING A SPECTRAL CLUSTERING BASED METHOD

The difference between different spectral clustering based algorithms is in the way that they compute the adjacency matrix. Accordingly, we should define proper metrics using which we could determine how accurate/useful is the estimated adjacency matrix. The authors of [Soltanolkotabi et al., 2012] used the number of false connections (any non-zero connection between two nodes/data-points while they belong to different clusters) as a metric to assess the estimated adjacency matrix. However, the graph clustering algorithms such as spectral clustering can yield an exact clustering of the data even if there are a significant amount of false connections in the estimated adjacency matrix provided that the estimated weights on the true connections are sufficiently stronger than the weights of the false connections. Therefore, in this paper, we use the following criteria to assess the quality of a adjacency matrix and we analyze the subspace clustering algorithms to reveal if/how they satisfy Requirement 1.

**Requirement 1.** *Suppose $\mathbf{A} \in \mathbb{R}^{M_2 \times M_2}$ is the estimated adjacency matrix. We require all the columns of $\mathbf{A}$ to satisfy*

$$\frac{\kappa}{m-1}\|\mathbf{a}_{i\mathcal{I}_i^\perp}\|_p^p < \|\mathbf{a}_{i\mathcal{I}_i}\|_p^p \, ,$$

*where $\mathcal{I}_i = \{j \ | \ k_i = k_j\}$, $\mathcal{I}_i^\perp = \{j \ | \ k_i \neq k_j\}$, $k_i = \arg\max_j \|\mathbf{U}_j^T\mathbf{d}_i\|_2$, and $\mathbf{a}_{i\mathcal{I}_i}$ contains the elements of $\mathbf{a}_i$ whose indexes are in $\mathcal{I}_i$.*

The parameter $\kappa$ is chosen greater than 1 and it determines how well the adjacency matrix represents the clustering structure of the data. Evidently, the higher is $\kappa$, the more challenging it is for a subspace clustering algorithm to satisfy Requirement 1. In the following sections, we discuss the role of parameter $p$ and we analyze MFC/iPursuit such that they satisfy Requirement 1 with $p = 1/p = 2$.

**Remark 1.** *Even if $\mathbf{A}$ satisfies Requirement 1 with a large $\kappa$, it does not necessarily mean that Spectral Clustering yields exact clustering. Similarly, proving that $\mathbf{A}$ does not contain any false connection (as in [Soltanolkotabi et al., 2012]) also does not guarantee exact clustering. However, these measures are useful to assess how clear the estimated $\mathbf{A}$ represents the clustering structure. In addition, although Requirement 1 does not guarantee exact clustering by the spectral clustering step, it is very similar to the sufficient*

*condition stated in [Ling and Strohmer, 2020] to guarantee that the spectral clustering algorithm yields the exact clustering. Specifically, [Ling and Strohmer, 2020] proves that if*

$$\max_i \|\mathbf{a}_{i\mathcal{I}_i^\perp}\|_1 < \frac{\min_k \gamma_2(\mathcal{L}(A_k))}{4} \,,$$

*then the spectral clustering algorithm studied in [Ling and Strohmer, 2020] yields an exact clustering where $\gamma_2(\mathcal{L}(A_k))$ is the second smallest eigenvalue of graph Laplacian w.r.t. the $k^{th}$ cluster and $\mathbf{A}_k \in \mathbb{R}^{n \times n}$.*

# 4 THEORETICAL STUDIES

This section focuses on analyzing MFC/iPursuit and revealing the key factors in its performance. First, we discuss the underlying connection between iPursuit and MFC and this interesting connection is utilized to analyze both algorithms using similar techniques. We refer the reader to [Rahmani, 2022] for the proofs of all the presented results. In the following sections, we utilize the parameters defined bellow.

**Definition 1.** *Suppose $\mathbf{D}$ follows Data Model 1. We define $\Delta_{\min} = \min_j \{ \inf_{\substack{\|\mathbf{u}\|=1 \\ \mathbf{u} \in \mathcal{S}_j}} \|\mathbf{u}^T \mathbf{D}_j\|_p^p \}_{j=1}^m$, $\dot{\Delta}_{\max} = \max_j \{ \sup_{\substack{\|\mathbf{u}\|=1 \\ \mathbf{u} \in \mathbb{R}^{r-s}}} \|\mathbf{u}^T \dot{\mathbf{D}}_j\|_p^p \}_{j=1}^m$, $\bar{\Delta}_{\max} = \max_j \{ \sup_{\substack{\|\mathbf{u}\|=1 \\ \mathbf{u} \in \mathbb{R}^s}} \|\mathbf{u}^T \bar{\mathbf{D}}_j\|_p^p \}_{i=1}^m$ ,and $\phi = \max_{j \neq t} \|\dot{\mathbf{U}}_t^T \dot{\mathbf{U}}_j\|$. In addition, when $y > x$, we define $\sigma_l(\frac{x}{y}, \delta) = \frac{x - 2\sqrt{x \log \frac{2M_2}{\delta}}}{y + 2\sqrt{(y-x) \log \frac{2M_2}{\delta}} + 2 \log \frac{2M_2}{\delta} - 2\sqrt{x \log \frac{2M_2}{\delta}}}$ and $\sigma_u(\frac{x}{y}, \delta) = \frac{x + 2\sqrt{x \log \frac{2M_2}{\delta}} + 2 \log \frac{2M_2}{\delta}}{y + 2\sqrt{x \log \frac{2M_2}{\delta}} + 2 \log \frac{2M_2}{\delta} - 2\sqrt{(y-x) \log \frac{2M_2}{\delta}}}$.*

The parameters $\Delta_{\min}$, $\dot{\Delta}_{\max}$, and $\bar{\Delta}_{\max}$ are similar to permeance statistic [Lerman et al., 2015] which indicates how well the data points are distributed inside the subspaces. For instance, when the columns of $\mathbf{D}_i$ in $\mathcal{S}_i$ are concentrated around a direction, the value of $\inf_{\substack{\|\mathbf{u}\|=1 \\ \mathbf{u} \in \mathcal{S}_i}} \|\mathbf{u}^T \mathbf{D}_i\|_p^p$ is small in comparison to when the data points are uniformly distributed in $\mathcal{S}_i$. Although the permeance statistic appears in the presented results, it does not necessarily mean that iPursuit and MFC require a uniform distribution of data pints inside the subspaces and the reason that it appears is that the sufficient conditions guarantee the performance under the worst case scenarios. The parameter $\phi$ indicates how close the innovative components $\{\dot{\mathcal{S}}_i\}_{i=1}^m$ are to each other.

**Remark 2.** *It is important to note that $\phi$ only measures the affinity between the innovative components $\{\dot{\mathcal{S}}_i\}_{i=1}^m$. In other word, even if two subspaces $\mathcal{S}_i$ and $\mathcal{S}_j$ heavily intersect such that $\|\mathbf{U}_i^T \mathbf{U}_j\|_\sigma$ is nearly equal to 1, $\|\dot{\mathbf{U}}_i^T \dot{\mathbf{U}}_j\|$ could be small if the innovative components are incoherent with each other. In the following results, it is shown*

*that in contrast to most of subspace segmentation methods whose performance depend on $\max_{j \neq t} \|\mathbf{U}_t^T \mathbf{U}_j\|_\sigma$, the performance of iPursuit and MFC mainly depends on the distance between the innovative components.*

## 4.1 THE CONNECTION BETWEEN IPURSUIT AND MFC

The cost function of iPursuit (2) encourages the optimal direction $\mathbf{c}_i^*$ to be orthogonal to the maximum number of data points. If the innovation assumption (Assumption 1) holds and $\mathbf{c}_i^* \in \vec{\mathcal{S}}_{k_i}$ for all the data points, then $\mathbf{A} = |\mathbf{D}^T \mathbf{C}^*|$ does not include any false connection. However, in practice the innovation assumption is not essential and $\mathbf{A} = |\mathbf{D}^T \mathbf{C}^*|$ can yield an accurate clustering of the data even if $|\mathbf{D}^T \mathbf{C}^*|$ is not a sparse matrix [Rahmani and Atia, 2017c, Ling and Strohmer, 2020]. A direct conclusion is that it may not be essential to employ $\ell_1$-norm in the cost function of (2). Accordingly, in this section, we investigate an iPursuit algorithm whose $i^{th}$ optimal direction is obtained as the optimal point of

$$\min_{\mathbf{c}} \|\mathbf{c}^T \mathbf{D}\|_2 \quad \text{subject to} \quad \mathbf{c}^T \mathbf{d}_i = 1 \,. \quad (3)$$

The following lemma shows that the iPursuit algorithm which employs $\ell_2$-norm to compute the optimal directions is equivalent to MFC.

**Lemma 1.** *Define $\mathbf{C}^*$ as the optimal point of*

$$\min_{\mathbf{C}} \|\mathbf{D}^T \mathbf{C}\|_{2,1} \quad \text{subject to} \quad \text{diag}(\mathbf{C}^T \mathbf{D}) = \mathbf{1} \,,$$

*and define $\mathbf{A} = |\mathbf{D}^T \mathbf{C}^*|$. Then $\mathbf{A}(i,j) = \frac{|\mathbf{v}^{i^T} \mathbf{v}^j|}{\|\mathbf{v}^i\|_2^2}$ .*

Lemma 1 shows that iPursuit is equivalent to MFC when $\ell_2$-norm is employed to compute the optimal vectors (note that the denominator of $\mathbf{A}(i,j) = \frac{|\mathbf{v}^{i^T} \mathbf{v}^j|}{\|\mathbf{v}^i\|_2^2}$ is the same for all the entries of a row and Step 4 of Algorithm 2 normalizes the degree of the nodes and keeps the dominant entries of each row). We leverage this connection between MFC and iPursuit to provide an analysis which is applicable to both algorithms. In the following theoretical results, $p$ appears as a parameter in the sufficient conditions. If $p = 1$, the sufficient condition corresponds to iPursuit and if $p = 2$, then the sufficient condition corresponds to MFC.

## 4.2 AN ANALYSIS FOR MFC AND IPURSUIT

The following theorem provides a sufficient condition to guarantee that Requirement 1 is satisfied. The presented results are applicable to both iPursuit and MFC since it is assumed that $\mathbf{A} = |\mathbf{D}^T \mathbf{C}^*|$ where $\mathbf{C}^*$ is obtained via solving

$$\min_{\mathbf{C}} \|\mathbf{D}^T \mathbf{C}\|_{p,1} \quad \text{subject to} \quad \text{diag}(\mathbf{C}^T \mathbf{D}) = \mathbf{1} \,. \quad (4)$$

**Theorem 2.** *Suppose that* $\mathbf{D}$ *follows Data Model 1 and* $\mathbf{A} = |\mathbf{D}^T \mathbf{C}^*|$ *where* $\mathbf{C}^*$ *is the optimal point of (4). If*

$$\min_i \frac{\|\beta_i\|_2^p}{\|\mathbf{d}_i\|_2^p} \Delta_{\min} \geq$$
$$\dot{\Delta}_{\max} \kappa \left( \frac{1}{\kappa + (m-1)} + \frac{m-1}{\kappa + (m-1)} \phi^p \right), \tag{5}$$

*then* $\mathbf{A}$ *satisfies Requirement 1.*

In contrast to former theoretical results which require $\max_{j \neq t} \|\mathbf{U}_t^T \mathbf{U}_j\|_\sigma$ to be sufficiently small, the presented guarantee is concerned with $\max_{j \neq t} \|\dot{\mathbf{U}}_t^T \dot{\mathbf{U}}_j\|$ and note that $\max_{j \neq t} \|\dot{\mathbf{U}}_t^T \dot{\mathbf{U}}_j\|$ can stay small even if the subspaces have a high dimension of intersection (i.e., $\|\mathbf{U}_t^T \mathbf{U}_j\|_\sigma$ is nearly equal to 1). When $m$, the number of clusters, is large, the sufficient condition can be roughly simplified into $\phi^p \leq \frac{\Delta_{\min}}{\kappa \dot{\Delta}_{\max}} \min_i \frac{\|\beta_i\|_2^p}{\|\mathbf{d}_i\|_2^p}$, which means that the higher is the dimension of intersection, the more distanced the innovative components should be. The sufficient condition requires all the data points to have a sufficiently strong projection on the innovative component.

### 4.3 PROBABILISTIC GUARANTEES

In this section, we simplify the result presented in Theorem 2 in two steps. First, we presume a random model for the distribution of the data points and in the second step, we consider a random model for the generation of the subspaces. We start with the first step as follows.

**Assumption 2.** *Each matrix* $\mathbf{D}_i \in \mathbb{R}^{M_1 \times n}$ *is generated as* $\mathbf{D}_i = \mathbf{U}_i \mathbf{G}_i$ *where the elements of* $\mathbf{G}_i \in \mathbb{R}^{r \times n}$ *are sampled independently from* $\mathcal{N}(0, \frac{1}{\sqrt{r}})$.

Assumption 2 ensures that the distribution of $\frac{\mathbf{d}_i}{\|\mathbf{d}_i\|_2}$ is uniformly at random on $\mathbb{S}^{M_1 - 1} \cap \mathcal{S}_{k_i}$. Note that $\mathbb{E}[\|\mathbf{d}_i\|_2^2] = 1$ and in the following theorems, we do not normalize the $\ell_2$-norm of the data points to make the analysis easier. In this section, we derive the guarantees for $p = 2$ and similar guarantees for $p = 1$ can be established.

**Theorem 3.** *Suppose* $\mathbf{D}$ *follows Data Model 1, matrices* $\{\mathbf{D}_i\}_{i=1}^m$ *are generated as in Assumption 2, and adjacency matrix* $\mathbf{A}$ *is computed as in Theorem 2 with* $p = 2$. *If*

$$(\frac{n}{r} - \eta_{\delta_r})\sigma_l \left( \frac{r-s}{r}, \delta \right) \geq$$
$$\kappa(\frac{1}{\kappa + m - 1} + \phi^2 \frac{m-1}{\kappa + m - 1})\left( \frac{n}{r} + \frac{r-s}{r}\eta_{\delta_{r-s}} \right) \tag{6}$$

*where* $\eta_{\delta_x} = \max(\frac{4z_{\delta_x}}{3} \log \frac{2\,x\,m}{\delta}, \sqrt{4\frac{n(x+3)}{x^2} \log \frac{2xm}{\delta}})$ *and* $z_{\delta_x} = 1 + 2\sqrt{\frac{1}{x} \log \frac{2nm}{\delta}} + \frac{2}{x} \log \frac{2nm}{\delta}$, *then Requirement 1 with* $p = 2$ *is satisfied with probability at least* $1 - 5\delta$.

Theorem 3 reveals several interesting points about the requirements of the algorithms. First it confirms our intuition about the relation between the dimension of subspaces and the required number of data points. The sufficient condition states that $n/r$ should be sufficiently large to ensure that Requirement 1 is satisfied. When $n/r$ is sufficiently large, then $(\frac{n}{r} - \eta_{\delta_r})$ is nearly equal to $n/r$. Therefore, when $m$ is large, the sufficient condition roughly states that $\phi^2$ should be sufficiently smaller than $\frac{1}{\kappa} \frac{r-s}{r}$. In other word, Theorem 3 clearly indicates that the higher is the dimension of intersection, the more separable their innovative components should be. Next, we further simplify the sufficient condition via assuming a random model for the distribution of subspaces.

**Theorem 4.** *Suppose* $\mathbf{D}$ *and* $\mathbf{A}$ *are generated as in Theorem 3 and* $\{\dot{\mathcal{S}}_i\}_{i=1}^m$ *and* $\mathcal{S}$ *are chosen independently and uniformly at random. If*

$$\left( \frac{n}{r} - \eta_{\delta_r} \right) \sigma_l \left( \frac{r-s}{r}, \delta \right) \geq$$
$$\kappa \left( \frac{n}{r} + \frac{r-s}{r}\eta_{\delta_{r-s}} \right) \left( \frac{1}{\kappa + m - 1} + \frac{c_\delta(r-s)^2}{M_1} \frac{m-1}{\kappa + m - 1} \right)$$

*then Requirement 1 is satisfied with probability at least* $1 - 6\delta$, *where* $c_\delta = 3\max\left( 1, \sqrt{\frac{8M_1\pi}{(M_1-1)(r-s)}}, \sqrt{\frac{16M_1 \log \frac{mr}{\delta}}{(M_1-1)(r-s)}} \right)$.

If we simplify the sufficient condition, Theorem 4 roughly states that $M_1$ should be sufficiently larger than $\kappa r(r - s)\sqrt{\log m}$. The main reason is that the subspaces and their innovative components are generated uniformly at random and the higher is the dimension of the ambient space, the less coherent they are in expectation.

**Remark 3.** *The main purpose of the presented analysis is to demonstrate the key performance factors of the MFC/iPursuit algorithms and to show why they are notably robust to the strong intersection between the span of clusters. If we want to go further and use the theoretical results to compare MFC/iPursuit against the other subspace clustering algorithms, we need to analyze the other methods using the utilized criteria (Requirement 1). Although it goes beyond the scope of this paper, Section 4.5 presents a full analysis of the TSC algorithm based on Requirement 1 to show why MFC can strongly outperform TSC while their computation complexities are not much different.*

### 4.4 WITH THE INNOVATION ASSUMPTION

The innovation assumption (Assumption 1) is not essential in the performance of MFC/iPursuit and we did not use it in any of the presented studies. However, the innovation assumption can be utilized to establish stronger guarantees. In this section, two theorems are presented whose only difference with Theorem 2 and Theorem 4 is that they assume that Assumption 3 (stated bellow) holds.

**Assumption 3.** *It is assumed that* $\mathbf{D}$ *follows Data Model 1 and* $\dim(\dot{\mathcal{S}}_i \cap \mathcal{P}_i) = 0$ *where* $\mathcal{P}_i = \oplus_{k \neq i} \mathcal{S}_i$.

Assumption 3 ensures that each innovative component $\dot{\mathcal{S}}_i$ is independent from the direct sum of all the other subspaces. The following theorem presumes that Assumption 3 holds.

**Theorem 5.** *Suppose* $\mathbf{D}$ *follows Assumption 3, define* $\vec{\mathcal{S}}_i$ *as the column space of* $(\mathbf{I} - \mathbf{P}_i\mathbf{P}_i^T)\mathbf{U}_i$, *define* $\vec{\mathbf{U}}_i$ *as a basis for* $\vec{\mathcal{S}}_i$, *and assume* $\mathbf{A} = |\mathbf{D}^T\mathbf{C}^*|$ *where* $\mathbf{C}^*$ *is the optimal point of (4). If*

$$\min_i \frac{\|\beta_i\|_2^p}{\|\mathbf{d}_i\|_2^p} \ \min_i \|\vec{\mathbf{U}}_{k_i}^T \dot{\mathbf{U}}_{k_i}\|_m^p \geq \frac{\kappa}{\kappa + m - 1} \frac{\dot{\Delta}_{\max}}{\Delta_{\min}} \ , \quad (7)$$

*then Requirement 1 is satisfied where* $\|\vec{\mathbf{U}}_{k_i}^T \dot{\mathbf{U}}_{k_i}\|_m$ *is the minimum singular value of* $\vec{\mathbf{U}}_{k_i}^T \dot{\mathbf{U}}_{k_i}$.

The subspace $\vec{\mathcal{S}}_i$ was defined as the projection of $\mathcal{S}_i$ onto $(\oplus_{k \neq i}\mathcal{S}_k)^\perp$ which is equivalent to the projection of $\dot{\mathcal{S}}_i$ onto $(\oplus_{k \neq i}\mathcal{S}_k)^\perp$. The closer is $\dot{\mathcal{S}}_i$ to $\vec{\mathcal{S}}_i$, the more incoherent is $\dot{\mathcal{S}}_i$ with the innovative component of the other clusters since $\vec{\mathcal{S}}_i$ is orthogonal to $\oplus_{j \neq i}\dot{\mathcal{S}}_j$. This is the reason we have $\|\vec{\mathbf{U}}_{k_i}^T \dot{\mathbf{U}}_{k_i}\|_m$ on the LHS of (7) because

$$\|\vec{\mathbf{U}}_{k_i}^T \dot{\mathbf{U}}_{k_i}\|_m = \min_{\|u\|_2 = 1} \|\vec{\mathbf{U}}_{k_i}^T \dot{\mathbf{U}}_{k_i}\mathbf{u}\|_2$$

is a measure of coherence between $\dot{\mathcal{S}}_i$ and $\vec{\mathcal{S}}_i$. Therefore, similar to Theorem 2, Theorem 5 states that the weaker is the projection of data points onto the innovative components, the more distanced the innovative components should be. The major difference between the condition of Theorem 2 and that of Theorem 5 is that in (7) $m$ plays a stronger role and (7) states that increasing $m$ (provided that it does not increase the coherency between the innovative components) can enhance the chance of MFC/iPursuit to satisfy Requirement 1. The following theorem provides a more explicit sufficient condition via assuming the random data model used in Theorem 4.

**Theorem 6.** *Suppose* $\mathbf{D}$ *and* $\mathbf{A}$ *are generated as in Theorem 4 and assume that* $M_1 > s + (r - s)m$. *If*

$$\sigma_l\left(\frac{r-s}{r}, \delta\right) \sigma_l\left(\frac{\vartheta}{M_1}, \delta\right) \geq$$

$$\frac{\kappa}{\kappa + m - 1} \frac{\frac{n}{r} + \frac{r-s}{r}\eta_{\delta\,r-s}}{\frac{n}{r} - \eta_{\delta\,r}} \quad (8)$$

*where* $\vartheta = M_1 - \big(s + (r - s)(m - 1)\big)$, *then Requirement 1 with* $p = 2$ *is satisfied with probability at least* $1 - 6\delta - \epsilon$ *where* $\epsilon$ *is the probability that the rank of* $\mathbf{D}$ *is less* $s + (r - s)m$.

Note that Theorem 6 does not need to explicitly presume that Assumption 3 holds because when $M_1 > s + (r - s)m$,

Assumption 3 is satisfied with an overwhelming probability [Vershynin, 2010]. The sufficient condition roughly states that when $n/r$ is large enough, then $\frac{r-s}{s}\frac{\vartheta}{M_1}$ should be sufficiently larger than $\frac{\kappa}{\kappa+m}$ to guarantee that the requirement is satisfied with high probability. The value of $\frac{\vartheta}{M_1}$ increases when $M_1$ increases and it converges to 1 when $r_d/M_1$ decreases.

Theorem 2, Theorem 5, and Theorem 6 indicate that if $\mathbf{D}$ follows Data Model 1, then the larger is the number of clusters, the more likely it is for MFC/iPursuit to satisfy Requirement 1 provided that increasing $m$ does not increase the coherency between $\{\dot{\mathcal{S}}_i\}_{i=1}^m$. This fact might sound counter intuitive, but it is an accurate prediction. For instance, suppose that $\mathbf{D}$ is generated as in Theorem 6, the first $n$ columns of $\mathbf{D}$ lie in $\mathcal{S}_1$, $n = 200$, $r = 10$, $s = 8$, and $M_1 = 400$. Define

$$\mathbf{a}_{\mathcal{S}_1} = \frac{1}{n}\sum_{i=1}^n \mathbf{a}_i$$

where $\mathbf{a}_i$ is the $i^{th}$ column of $\mathbf{A}$. Therefore, $\mathbf{a}_{\mathcal{S}_1}$ is the average of the first $n$ columns of $\mathbf{A}$ which are corresponding to data points in $\mathcal{S}_1$. Figure 1 shows $\mathbf{a}_{\mathcal{S}_1}$ with different values of $m$ for the adjacency matrices computed by MFC and the TSC algorithm [Heckel and Bölcskei, 2013] which computes $\mathbf{A} = |\mathbf{D}^T\mathbf{D}|$. Ideally, we should observe that the expected value of the elements of $\mathbf{a}_{\mathcal{S}_1}[1:n]$ are sufficiently larger than the expected value of the elements of $\mathbf{a}_{\mathcal{S}_1}[n:M_2]$. One can observe that when $\mathbf{A} = |\mathbf{D}^T\mathbf{D}|$, the elements of $\mathbf{a}_{\mathcal{S}_1}[1:n]$ are not much distinguishable from the elements of $\mathbf{a}_{\mathcal{S}_1}[n:M_2]$ with both $m = 2$ and $m = 10$. In contrast, when $\mathbf{A} = |\mathbf{D}^T\mathbf{C}^*|$ and when $m = 10$, $\|\mathbf{a}_{\mathcal{S}_1}[1:n]\|_2$ is clearly larger than $\frac{1}{m-1}\|\mathbf{a}_{\mathcal{S}_1}[n:M_2]\|_2$. The last plot of Figure 1 shows the effect of $m$ on the quality of the computed adjacency matrix in a more clear way. Define parameter $\hat{\kappa}$ as follows

$$\kappa' = \frac{(m - 1)\ \|\mathbf{a}_{\mathcal{S}_1}[1:n]\|_2^2}{\|\mathbf{a}_{\mathcal{S}_1}[n:M_2]\|_2^2}\ . \quad (9)$$

Parameter $\kappa'$ shows how clear the adjacency matrix separates the data points in $\mathcal{S}_1$ from the other clusters. The last plot (first from right), shows $\kappa'$ versus $m$ for both MFC and TSC. One can observe that $\kappa'$ notably increases as $m$ increases when $\mathbf{A} = |\mathbf{D}^T\mathbf{C}^*|$ which means that the quality of the estimated adjacency matrix improves as $m$ increases. In sharp contrast, increasing $m$ does not show a positive/negative impact on the computed adjacency matrix by Algorithm 3.

It is important to note that the conclusion that the performance of MFC/iPursuit improves if $m$ increases is not a general rule. When $M_1$ is not sufficiently large, as $m$ increases, the distance between the subspaces (and the distance between their innovative components) decreases and it degrades the performance of the algorithms. Moreover, the reason that in Theorem 4 and Theorem 6 the coherency

between the subspaces decreases as $M_1$ increases is due to the presumed model for the generation of the subspaces and it is not a general rule that $\phi$ decreases as $M_1$ increases.

---

**Algorithm 3** Inner-Product based Subspace Clustering [Heckel and Bölcskei, 2013] (TSC Algorithm)

**Input.** The input is data matrix $\mathbf{D} \in \mathbb{R}^{M_1 \times M_2}$.

**1. Data Preprocessing.** Normalize the $\ell_2$-norm of the columns of $\mathbf{D}$, i.e., set $\mathbf{d}_i$ equal to $\mathbf{d}_i/\|\mathbf{d}_i\|_2$ for all $1 \leq i \leq M_2$.

**2.** Define $\mathbf{A} = |\mathbf{D}^T\mathbf{D}|$.

**3.** Similar to Step 4 in Algorithm 1.

**4.** Similar to Step 5 in Algorithm 1.

**Output:** The identified clusters.

---

## 4.5 COMPARISON WITH THE TSC ALGORITHM

In this section, we theoretically compare the TSC algorithm against against MFC/iPursuit. Both MFC/iPursuit and Algorithm 3 use inner-product as the kernel function to measure the similarity between data points. However, in sharp contrast to Algorithm 3, MFC/iPursuit computes the inner-product between the directions of innovation and the data points as opposed to computing the inner-product between the data points. In [Rahmani and Atia, 2017b,c] and in this paper, it is shown that this difference makes MFC/iPursuit able to notably outperform TSC in most of scenarios. In order to clarify the reason behind this performance difference, we provide similar analysis for Algorithm 3 and we compare the requirements of MFC/iPursuit against those of Algorithm 3. Although the presented theorems only include sufficient conditions (not necessary conditions), their comparison is insightful.

**Theorem 7.** *Suppose $\mathbf{D}$ follows Data Model 1. If*

$$1 \geq \kappa \max_i \left\{ \frac{\|\alpha_i\|_2^p}{\|\mathbf{d}_i\|_2^p} \right\} \frac{\bar{\Delta}_{\max}}{\Delta_{\min}} + \\ \kappa \max_i \left\{ \frac{\|\beta_i\|_2^p}{\|\mathbf{d}_i\|_2^p} \right\} \phi^p \frac{\dot{\Delta}_{\max}}{\Delta_{\min}} , \quad (10)$$

*then $\mathbf{A} = |\mathbf{D}^T\mathbf{D}|$ satisfies Requirement (1).*

There are two terms on the RHS of the sufficient condition where only the second term is weighted by $\phi$. Even in the best case scenario where the innovative components are orthogonal to each other, i.e., $\phi = 0$, it may not be possible to satisfy the sufficient condition. For instance, suppose $s/r$ is nearly equal to one and assume that the elements of $\beta_i$ and $\alpha_i$ are sampled independently from $\mathcal{N}(0,1)$. In this scenario, $\mathbb{E}\left[\frac{\|\alpha_i\|_2^2}{\|\mathbf{d}_i\|_2^2}\right] = \frac{s}{m} \approx 1$ and it may not be possible to satisfy the sufficient condition even for $\kappa = 2$. The main reason is that when $s/m$ is high, the inner-product value between data points in different clusters are high, no matter how well separated the innovative components are. In sharp

contrast to Algorithm 3, MFC/iPursuit utilize the inner-product between the optimal directions and the data points to construct the adjacency matrix and when $s/m$ is high, the optimal directions are strongly incoherent with $\mathcal{S}$ and this feature makes the role of the innovative components notably more significant. In order to make a more explicit comparison, we derive the sufficient condition for Algorithm 3 while it is assumed that the data is generated as in Theorem 4. The following theorem provides the result.

**Theorem 8.** *Suppose $\mathbf{D}$ is generated as in Theorem 4 and $\mathbf{A} = |\mathbf{D}^T\mathbf{D}|$. If*

$$\left( \frac{n}{r} - \eta_{\delta_r} \right) \geq \kappa\, \sigma_u \left( \frac{s}{r}, \delta \right) \left( \frac{n}{r} + \frac{s}{r}\, \eta_{\delta_s} \right) + \\ \kappa\, \sigma_u \left( \frac{r-s}{r}, \delta \right) \left( \frac{n}{r} + \frac{r-s}{r}\, \eta_{\delta_{r-s}} \right) \left( \frac{c_\delta (r-s)^2}{M_1} \right) ,$$

*then Requirement 1 with $p = 2$ is satisfied with probability at least $1 - 9\delta$, where $c_\delta$ was defined in Theorem 4 and $\eta_{\delta_x}$ was defined in Theorem 3.*

The first term on the RHS of the sufficient condition of Theorem 8 is the dominant term when $s$ is large. When there are a sufficiently large number of data points in the clusters ($n/r$ is large enough), the sufficient condition roughly states that $\kappa \frac{s}{r}$ should be sufficiently smaller than 1. However, it is not feasible to satisfy this condition in many scenarios. For instance, if we choose $\kappa = 2$, then the sufficient condition can be satisfied only when $s/r > 0.5$.

In summary, comparing the sufficient conditions suggests that in sharp contrast to Algorithm 3 which fails when the span of clusters are close, MFC/iPursuit can effectively leverage the innovative components of the clusters and if these innovative components are sufficiently separable ($\phi$ is sufficiently small), MFC/iPursuit might successfully distinguish the clusters.

## 5 NUMERICAL EXPERIMENTS

This paper does not present a new clustering algorithm and the main focus was to provide a deep understating and analysis of the MFC/iPursuit algorithms. We refer the reader to [Kanatani, 2001, Costeira and Kanade, 1998, Boult and Brown, 1991, Vidal, 2011, Rahmani and Atia, 2017a,b] for numerical studies of the MFC/iPursuit algorithms. The focus of the presented experiments are to demonstrate some of the features of the algorithms which was predicted by the presented theoretical studies. For iPursuit, MFC, and TSC, the graph preprocessing step (Step 4 in Algorithm 1) was done as follows. For each column of $\mathbf{A}$, 8 largest elements were kept and the rest of elements were set to zero. Clustering error is defined as $\frac{N_e}{M_2}$ where $N_e$ is the total number of misclassified data points. In the appendix, we have included a simple numerical experiment showing that exact clustering can be achieved if Requirement 1 holds even for small values of $\kappa$.

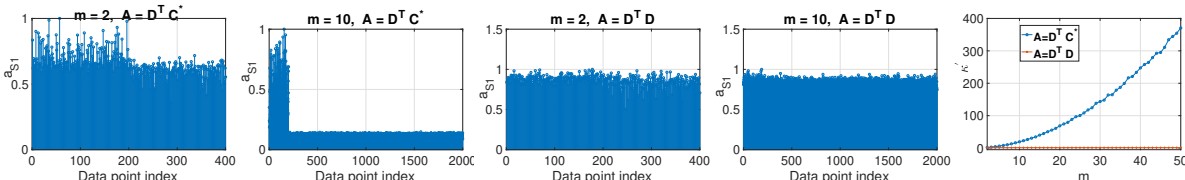

Figure 1: The first 4 plots (from LHS) show the elements of $\mathbf{a}_{\mathcal{S}_1} = \frac{1}{n}\sum_{i=1}^{n}\mathbf{a}_i$ with different number of clusters for MFC and Algorithm 3. The first $n = 200$ data points lie in first cluster, $r = 10$, $s = 8$, and $M_1 = 400$. The last plot demonstrates parameter $\kappa'$ defined in (9) versus $m$. One can observe that in this experiment increasing $m$ improves the quality of the adjacency matrix computed by MFC.

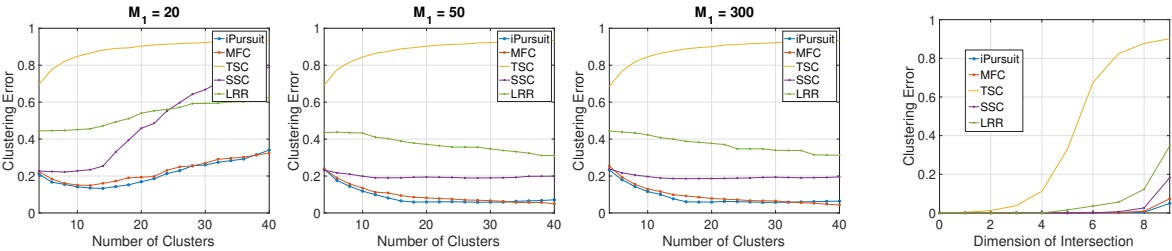

Figure 2: First three plots from left: Clustering error versus the number of clusters for different values of $M_1$ where $r = 10$, $s = 9$, and $n = 100$. First plot from right: This plot demonstrates clustering error versus $s$. In this experiment, $M_1 = 40$, $r = 10$, and $n = 100$.

## 5.1 THE DIMENSION OF INTERSECTION BETWEEN THE SUBSPACES

In the presented deterministic results (Theorem 2 and Theorem 5), we observed that $\frac{\|\beta_i\|_2}{\|\mathbf{d}_i\|_2}$ is an important factor in the performance of MFC/iPursuit and in the probabilistic results, this factor appeared as $\frac{r-s}{r}$. The purpose of this experiment is twofold. Firstly, we show that the accuracy of MFC/iPursuit degrades as $s$ increases (since $\frac{r-s}{r}$ decreases). Secondly, it is shown that MFC/iPursuit are notably robust against intersection between the span of clusters comparing to most of other methods. The first plot (from right) in Figure 2 shows clustering error versus $s$ where in this experiment $M_1 = 40$, $r = 10$, and $n = 100$ (the number of evaluation runs was 50). One can observe that the accuracy of MFC/iPursuit degrades as $s$ increases. However, both of them notably outperform the other methods when $s$ is high. The main reason is that as the presented theoretical studies indicated, the performance of MFC/iPursuit mainly depends on the coherency between the innovative components $\{\dot{\mathcal{S}}_i\}_{i=1}^{m}$ while most of other algorithms such as TSC require the span of clusters $\{\mathcal{S}_i\}_{i=1}^{m}$ to be sufficiently incoherent.

## 5.2 NUMBER OF CLUSTERS

In the theoretical results (Theorem 2 and Theorem 5), it was shown that the quality of the adjacency matrix computed by MFC/iPursuit might improve when $m$ increases. Specifically, the theoretical results suggested that when data follows Data Model 1 and as long as increasing $m$ does not increase the coherency between $\{\dot{\mathcal{S}}_i\}_{i=1}^{m}$, MFC/iPursuit can

yield a better adjacency matrix (an adjacency matrix with higher $\min_i \frac{(m-1)\|\mathbf{a}_{i\mathcal{I}_i}\|_P^P}{\|\mathbf{a}_{i\mathcal{I}_i^{\perp}}\|_P^P}$) if $m$ increases.

The first three plots (from left) in Figure 2 shows clustering error versus $m$ for different values of $M_1$ where in this experiment $r = 10$, $s = 9$, and $n = 100$ (the number of evaluation runs was 50). One can observe that when $M_1 = 300$ and when $M_1 = 50$, the accuracy of MFC/iPursuit improves when $m$ increases while when $M_1 = 20$, the accuracy degrades. The reason for this observation is that as the theoretical results indicated, both the number of clusters and the coherency between the innovative components contribute to the performance of the algorithms. When $M_1$ is not sufficiently large, increasing $m$ increases the coherency between the innovative components and it degrades the performances of the algorithms.

## 5.3 FACE CLUSTERING

In this experiment, we use the Extended Yale B dataset which contains 64 images for each of 38 individuals in frontal view and different illumination condition Lee et al. [2005]. In this dataset, since all the images were taken from the same frontal pose, the faces corresponding to each subject can be approximated with a low-dimensional subspace Basri and Jacobs [2003]. Thus, the images in this dataset can be modeled as a union of linear subspaces. In this experiment, we created $\mathbf{D}$ via vectorizing each image and using each image as a column of $\mathbf{D}$. To expedite the run-time, we projected the data on the span of the first 500 left singular

Table 1: Clustering error of different algorithms on the Extended Yale B dataset.

| Algorithm | iPursuit | MFC | LRR | SSC | TSC |
|---|---|---|---|---|---|
| Clustering error | 0.08 | 0.09 | 0.6 | 0.29 | 0.71 |

vectors. Define $\mathbf{s}$ as the vector of the singular values of $\mathbf{D}$ and define $\hat{\mathbf{s}} = \frac{\mathbf{s}}{\max_i \mathbf{s}(i)}$. In MFC, we estimated $r_d$ equal to the number of elements of $\hat{\mathbf{s}}$ which are greater than 0.01. Table 1 shows the clustering error of the clustering algorithms (number of misclassified data points divided by the total number of data points). One can observe that the performance of MFC and iPursuit are close to each other since they employ similar tools to build the adjacency matrix. In addition, they notably outperformed the other approaches and the main reason is that in this dataset the span of clusters are close to each other [Vidal, 2011]. The presented theoretical results indicated that MFC/iPursuit could yield a high quality adjacency matrix even if the span of clusters are close to each other because their performance mainly depend on the incoherency between the innovative components of the clusters.

## 5.4 REQUIREMENT 1

We discussed the fact that Requirement 1 indicates how clear the estimated adjacency matrix represents the clustering structure of the data and it is similar to the sufficient condition established in [Ling and Strohmer, 2020] which guarantees that the spectral clustering algorithm can yield exact clustering. In this experiment, we assume that $m = 4$ and $n = 100$ which means $M_2 = 400$. In order to construct $\mathbf{A}$, we sample each element of $\mathbf{A}$ from half-normal distribution and we normalize the elements such that $\frac{\kappa}{m-1} \|\mathbf{a}_{i\mathcal{I}_i^\perp}\|_1 = \|\mathbf{a}_{i\mathcal{I}_i}\|_1$ , for all $1 \leq i \leq M_2$. Figure 3 shows clustering error of the spectral clustering algorithm versus $\kappa$. One can observe that even a small value of $\kappa$ can guarantee exact clustering. Although the minimum value of $\kappa$ for which we can guarantee exact clustering depends on the distribution of the elements of $\mathbf{A}$, but it shows that (as the results in [Ling and Strohmer, 2020] suggests), exact clustering can be achieved if the false connections are sufficiently weaker than the true connections.

## 5.5 SUBSPACES WITH DIFFERENT INTERSECTIONS

In the presented theoretical studies, we utilized a single subspace $\mathcal{S}$ to model the intersection between the span of clusters to derive succinct sufficient conditions. In practice, every pair of clusters could have different intersecting subspaces. Define $\mathbf{S}_t \in \mathbb{R}^{M_1 \times 12}$ as a basis for a 12 dimensional subspace. In this experiment, we build the span of each clus-

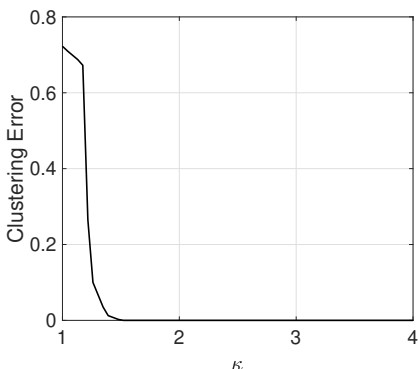

Figure 3: Clustering error versus parameter $\kappa$.

ter as

$$\mathbf{U}_i = [\mathbf{S}_i, \, \dot{\mathbf{U}}_i]$$

where the columns of $\mathbf{S}_i \in \mathbb{R}^{M_1 \times 9}$ are sampled from the columns of $\mathbf{S}_t$ randomly for each cluster. Therefore, each pair of clusters could have different intersecting subspaces. Figure 4 shows clustering error versus number of clusters where $M_1 = 60$. One can observe that the MFC algorithm yields accurate clustering of the data even if a single subspace does not model the intersection between all the pairs of subspaces.

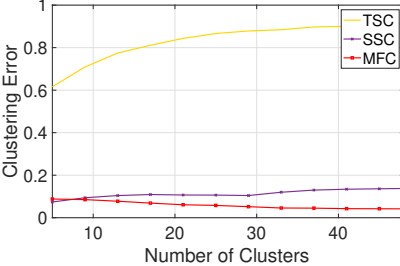

Figure 4: Clustering error versus number of clusters.

## CONCLUSION

It was shown that iPursuit is equivalent to a closed form matrix factorization based clustering algorithm if the direction search optimization problem is altered into a quadratic optimization problem. A novel analysis applicable to both algorithms were proposed which showed that in contrast to some of the other subspace clustering algorithms whose performance depend on the distance between the span of clusters, the performance of MFC/iPursuit mainly depends on the distance between the innovative components of the clusters.

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
