# OpenReview forum: "Clustering a Union of Linear Subspaces via Matrix Factorization and Innovation Search"
_auai.org/UAI/2022/Conference — UAI 2022 Oral_

### Official Review · Reviewer_dg3y · 2022-04-05

**Q2(1) Originality/Novelty:** 3
**Q2(2) Significance/Impact:** 2
**Q2(3) Correctness/Technical Quality:** 3
**Q2(6) Clarity Of Writing:** 4
**Q6 Overall Score:** 7
**Q8 Confidence In Your Score:** 4

**Q1 Summary And Contributions:**

Authors provide a theoretical analysis of Matrix Factorization based clustering (MCF) and Innovation Pursuit (iPursuit).

**Q2 Assessment Of The Paper:**

More detailed information regarding each of these aspects is given below:

**Q2(4) Quality Of Experiments (Optional):**

3: Good: The experimental evaluation is adequate, and the results convincingly support the main claims.

**Q2(5) Reproducibility:**

2: Fair: Key resources (e.g., proofs, code, data) are unavailable but key details (e.g., proof sketches, experimental setup) are sufficiently well-described for an expert to confidently reproduce the main results.

**Q3 Main Strengths:**

Authors first show the link/equivalence between MCF and iPursuit, and then provide a theoretical analysis: more precisely, they show that what they define as Requirement 1 is satisfied under reasonable assumptions. (Requirement 1 requires the computed weights in the adjacency matrix between clusters to be smaller than within clusters.)

**Q4 Main Weakness:**

From reading the paper, it is not awlays clear what are the assumptions under which MCF/iPursuit will work. For example, the Data Model 1 is a bit odd: all subspaces intersect in the same subspace S, all have the same dimension and same numner of data points. This is an unusual setting which should be discussed in more details. In particular, what would happen if some of these assumptions are violated?

**Q5 Detailed Comments To The Authors:**

I have one additional comment:
- Authors claim in the abstract that "MFC can outperform the other sophisticated subspace clustering methods in many challenging scenarios." but this is not so much discussed in the paper. In what challenging scenarios? What sophisticated subspace clustering methods?


**Q7 Justification For Your Score:**

The theoretical insignt for MCF and iPursuit are a nice contribution to the subspace clustering literature.

**Q9 Complying With Reviewing Instructions:**

1: Yes.

---

### Official Review · Reviewer_Qg7c · 2022-04-12

**Q2(1) Originality/Novelty:** 3
**Q2(2) Significance/Impact:** 3
**Q2(3) Correctness/Technical Quality:** 3
**Q2(6) Clarity Of Writing:** 3
**Q6 Overall Score:** 5
**Q8 Confidence In Your Score:** 2

**Q1 Summary And Contributions:**

This paper theoretically analyzes the Matrix Factorization based Clustering, and reveal the connection between MFC and the Innovation Pursuit algorithm. They show that the performance of MFC/iPursuit depends on the distance between the innovative components of the clusters.

**Q2 Assessment Of The Paper:**

More detailed information regarding each of these aspects is given below:

**Q2(4) Quality Of Experiments (Optional):**

3: Good: The experimental evaluation is adequate, and the results convincingly support the main claims.

**Q2(5) Reproducibility:**

3: Good: Key resources (e.g., proofs, code, data) are available and key details (e.g., proofs, experimental setup) are sufficiently well-described for competent researchers to confidently reproduce the main results.

**Q3 Main Strengths:**

1. The theoretical analysis of this work is solid and strong. They show that iPursuit is equivalent to MFC if the cost function is changed to a quadratic cost function while maintaining the theoretical results.
2. This paper firstly show that the performance of MFC/iPursuit depends on the distance between the innovative components of the clusters, which is surprising and interesting.

**Q4 Main Weakness:**

I would suggest the authors to add more numerical studies to analyze the performance of clustering methods. And I wonder how the real data obey the proposed Data Model 1. Maybe the authors can do some experiments on some real datasets.

**Q5 Detailed Comments To The Authors:**

See Q3 and Q4

**Q7 Justification For Your Score:**

This paper provides solid theoretical analysis on MFC and the findings are important to understand such clustering methods. And some numerical results are provided to validate the theoretical analysis. And I would suggest the authors to add more numerical studies on real datasets to analyze the performance of clustering methods as well as how well the data model 1 fits the real cases.

**Q9 Complying With Reviewing Instructions:**

1: Yes.

---

### Official Review · Reviewer_ZuFx · 2022-04-13

**Q2(1) Originality/Novelty:** 3
**Q2(2) Significance/Impact:** 3
**Q2(3) Correctness/Technical Quality:** 3
**Q2(6) Clarity Of Writing:** 3
**Q6 Overall Score:** 7
**Q8 Confidence In Your Score:** 3

**Q1 Summary And Contributions:**

The paper presents in-domain self-supervised pre-training strategies
which address the problem of specialized domains with a small corpus.
The paper also presents a heterogeneous graph convolution technique
with for multimodal sentiment analysis. The technique fuses
information from multiple sources to provide improved performance.
Experimental evaluation is provided to validate the proposed technique.

**Q2 Assessment Of The Paper:**

More detailed information regarding each of these aspects is given below:

**Q2(4) Quality Of Experiments (Optional):**

2: Fair: The experimental evaluation is weak: important baselines are missing, or the results do not adequately support the main claims.

**Q2(5) Reproducibility:**

3: Good: Key resources (e.g., proofs, code, data) are available and key details (e.g., proofs, experimental setup) are sufficiently well-described for competent researchers to confidently reproduce the main results.

**Q3 Main Strengths:**

Theoretical analysis that establishes a connection between MFC and
iPursuit.

**Q4 Main Weakness:**

Lack of experiments with very high dimensions and fewer data points

**Q5 Detailed Comments To The Authors:**

Overall, it is very interesting.  The title doesn't seem to describe
the paper well. Data model 1 is not well motivated. It'd be nice to
provide some real problems that are represented by data model 1.

**Q7 Justification For Your Score:**

It is a good work. The paper's strengths outweigh its weaknesses.

**Q9 Complying With Reviewing Instructions:**

1: Yes.

---

### Decision · Program_Chairs · 2022-05-15

**Decision:**

Accept (Oral)

**Comment:**

Meta Review: The paper studies the Matrix Factorization base Clustering method and establish its connection with Innovation Pursuit. Theoretical results are presented in support of both approaches. The AC fully agree with the reviewers that the paper makes some very exciting novel contributions. We strongly urge the authors to incorporate their feedback to the reviews into their revised manuscript. In particular, please include the new Extended Yale B experiments. Also, it would be great if the authors could update the title of their manuscript as they proposed in their feedback.